# A Deformation of a Mercury Droplet under Acceleration in an Annular Groove

**DOI:** 10.3390/bios10060061

**Published:** 2020-06-09

**Authors:** HanYang Xu, Yulong Zhao, Kai Zhang, Zixi Wang, Kyle Jiang

**Affiliations:** 1State Key Laboratory for Manufacturing Systems Engineering, Xi’an Jiaotong University, Xi’an 710049, China; xuhanyang@stu.xjtu.edu.cn (H.X.); zhangkai1995@stu.xjtu.edu.cn (K.Z.); 2Xi’an Satellite Control Center, Xi’an 710043, China; wongzx1023@163.com; 3School of Mechanical Engineering, University of Birmingham, Birmingham B15 2TT, UK; k.jiang@bham.ac.uk

**Keywords:** liquid droplet MEMS sensor, droplet deformation, level set method, acceleration

## Abstract

Microelectromechanical system (MEMS) liquid sensors may be used under large acceleration conditions. It is important to understand the deformation of the liquid droplets under acceleration for the design and applications of MEMS liquid sensors, as this will affect the performance of the sensors. This paper presents an investigation into the deformation of a mercury droplet in a liquid MEMS sensor under accelerations and reports the relationship between the deformation and the accelerations. The Laminar level set method was used in the numerical process. The geometric model consisted of a mercury droplet of 2 mm in diameter and an annular groove of 2.5 mm in width and 2.5 mm in height. The direction of the acceleration causing the droplet to deform is perpendicular to the direction of gravity. Fabrication and acceleration experiments were conducted. The deformation of the liquid was recorded using a high-speed camera. Both the simulation and experimental results show that the characteristic height of the droplets decreases as the acceleration increases. At an acceleration of 10 m/s^2^, the height of the droplet is reduced from 2 to 1.658 mm, and at 600 m/s^2^ the height is further reduced to 0.246 mm. The study finds that the droplet can deform into a flat shape but does not break even at 600 m/s^2^. Besides, the properties of the material in the domain surrounding the droplet and the contact angle also affect the deformation of the droplet. This work demonstrates the deformation of the liquid metal droplets under acceleration and provides the basis for the design of MEMS droplet acceleration sensors.

## 1. Introduction

Liquid droplets have been wildly used as a component in many MEMS sensors and actuators, such as the droplet touch sensors reported by Kim in 2017 [1], the ferrofluid droplet actuators reported by Bijarchi in 2020 [2], and the angle droplet sensors provided by Xu in 2019 [3]. Such droplet sensors are considered to have advantages in high sensitivity and accuracy [4,5]. Meanwhile, the liquid droplets are subject to high viscosity and surface tension, which may hold them against breaking. Understanding the deformation of the droplets will help designers to analyze the performance of the MEMS droplet sensors. Usually, the changes in the output signals of a liquid droplet sensor are caused by the deformation or movement of the droplets in a groove [6]. Hence, there is a need to investigate the deformation of droplets caused by acceleration.

Droplet deformation is modelled with two phases: the droplet itself and the domain surrounding it. Therefore, multiphase methods need to be applied in the numerical process to track the connection surface of each domain. The level set multiphase method is one of the popularly used methods in analyzing such multiphase fluids, proposed by Sussman et al in 1994 [7]. In comparison with the volume of fluid (VOF) multiphase method, the level set method can simplify the process of capturing topological changes caused by surface tension [8,9]. Previous research in the collision process of the droplets on flat surfaces using multiphase flow methods include the materials of the surface and the droplet [10], the initial velocity of the droplet [11], and the surface tension of the droplet [12]. Alain proposed five collision modes based on the initial conditions and material properties, including bouncing and coalescence after minor deformation [13]. In 2019, Liu studied the impact of metal tin dust on a spherical bead at the acceleration of gravity using numerical simulation and experiments [14]. The results show that increasing the initial temperature, velocity and size of tin dust can increase the spread area formed after impact. Regarding the movement of the droplet in a channel, Aboutalebi used numerical simulation to study the influence of magnetic field intensity and droplet size on the ferromagnetic droplet splitting process in a symmetrical T-junction, and the boundary between the split and non-split regions where the droplets flow in the micro-T junction is predicted [15]. However, the research on the deformation of the droplet in an annular groove under a large acceleration is rare. Whether the deformation of the droplet in the annular groove is split or flattened, and the relationship between the deformation and the initial boundary or material is worth investigating.

This paper presents a numerical and experimental study of the deformation of a mercury droplet in an annular groove at acceleration. The mercury was selected as the research object mainly because of the high surface tension and conductivity of the mercury. Besides, mercury is widely used in current MEMS liquid droplet sensors. The numerical model consists of the air domain and the mercury domain. The direction of the acceleration is perpendicular to the direction of gravity. The Laminar level set (LS) method was used in the study to show the profile of droplets during the deformation process. The relationship between the height of the droplet after deformation and the applied acceleration was studied. Then, experiments were conducted to verify the simulation results. The results of simulation and experiments both show that the droplet deforms under acceleration, and the characteristic height of droplet decreases with the increase in acceleration. Meanwhile, the droplet does not split, even when the acceleration reaches 600 m/s^2^.

This research work contributes to the research community of liquid droplet sensors in the following aspects: the deformation of droplets under the influence of acceleration was presented by using Laminar level set method; the deformation results in both the experiment and simulation show that the characteristic height of a droplet will decrease as the acceleration increases, which provides a basis for the dynamic design of the metal droplet acceleration sensors.

## 2. Computational Models and Methods

### 2.1. Geometric Models

Figure 1 shows the schematic of the initial static 3D model of a metallic liquid droplet in an annular groove. The computational fluid dynamics (CFD) model consists of both air and liquid droplet. The width of the groove is 2.5 mm, and the height is 2.5 mm. The material of the liquid droplet is mercury, which measures 2 mm in diameter. Here, the size of the annular groove is larger than the diameter of the droplet which makes the droplet easy to slide and improves the resolution of the sensor. The gravity is applied to the fluid in a negative Z-direction, and the acceleration in a negative Y-direction.

### 2.2. Governing Equations

During the deformation, the Reynold number does not reach the critical value of turbulence (less than 150). Therefore, the Navier–Stoke equation for incompressible laminar flow was used to solve the flow velocity and pressure change in the fluid flow [16,17]:(1)$$\rho \frac{\partial u}{\partial t}-\nabla \cdot \mu \left(\nabla u+{\left(\nabla u\right)}^{T}\right)+\rho \left(u\cdot \nabla \right)u+\nabla p=F_{st}+F$$
(2)∇⋅u=0
where **u** is the velocity vector; *ρ* is the density; *p* is the pressure; and *F* is the volume force, including gravity and body force caused by acceleration; *F_st_* is the surface tension. The Equation of *F* is as follows:(3)F=ρg+ρa
where *g* is the gravitational acceleration and *α* is the acceleration of the whole flow.

As the mercury droplet is surrounded by air, the model becomes a two-phase flow to count for both air and liquid droplet in simulation, and the change of the interfacing profile between the two phases needs to be considered and updated in simulation [18,19]. Hence, the LS method was used to solve the incompressible two-phase flow with surface tension model. The smoothing function of LS is used to describe the interface of the two phases, in which the LS function is always 0 in the air phase, 1 in the droplet phase, and from 0 to 1 in the interface of the two phases. Since the convection of LS function itself is smooth, it can replace the gradient of physical properties caused by convection at the interface [20]. In addition, a complex discontinuous tracking redraw grid algorithm is not required in LS method, which is based on the continuous approximation method. The surface tension and the local curvature of the interface are expressed as the volume force, which simplifies the capture topology changes caused by changes of surface tension in the calculation process [21]. In the model, the deformation of the interface is effectively tracked, and the variation of interface is captured by LS function. The LS function *ϕ* is given in Equation (4):(4)$$\frac{\partial \phi }{\partial t}+u\cdot \nabla \phi =0$$

The density and viscosity of the fluid are represented by the *ϕ* of the level set function as shown below:(5)$$\rho =\rho_{air}+H(\phi )(\rho_{mercury}-\rho_{air})$$
(6)$$\mu =\mu_{air}+H(\phi )(\mu_{mercury}-\mu_{air})$$
where *ρ_air_* and *μ_air_* are the density and viscosity of air domain, respectively; *ρ_mercury_* and *μ_mercury_* are the density and viscosity of mercury domain, respectively; and *H(ϕ)* is the Heaviside function, which is used to smooth approximation and avoid the problem of non-convergence caused by the Gibbs phenomenon. The value of the *H(ϕ)* is 1 in the droplet phase, 0 in the air phase, and a smooth transition at the interface of the two phases.

The unit normal vector of the interface **n**, and curvature *κ* of the interface of two-phase are shown as follows:(7)$$n=\frac{\nabla \phi }{\left|\nabla \phi \right|}$$
(8)$$\kappa =\nabla \cdot \frac{\nabla \phi }{\left|\nabla \phi \right|}$$

Here, the scalar surface tension can be expressed as a function of the curvature of the interface:(9)$$F_{st}=\sigma \kappa \delta \left(\phi \right)n$$
where *σ* is surface tension coefficient and *δ*(*ϕ*) is Dirac delta function.

It can be seen from Equations (1) and (3) that the total force in flow consists of surface tension, gravity, and the body force generated by acceleration. Therefore, the total forces of the fluid in the three directions of *X*, *Y*, and *Z* are shown below:(10)$$F_{x}=\sigma \kappa \frac{\partial \phi }{\partial x}\delta \left(\phi \right)$$
(11)$$F_{y}=\sigma \kappa \frac{\partial \phi }{\partial y}\delta \left(\phi \right)+a\rho$$
(12)$$F_{z}=\sigma \kappa \frac{\partial \phi }{\partial z}\delta \left(\phi \right)+\rho g$$

This numerical method requires both the incompressible Navier–Stokes equation module and the LS module for capturing the profile of the droplets in the two-phase flow [22]. Since the interface can be captured in LS function, it becomes feasible to study the change of the interface between the droplet and the air after the acceleration is applied.

### 2.3. Computational Models and Boundary Conditions

The numerical model was coded in COMSOL software, and 3D geometry was used in the model design. The model has accounted for the air and liquid phases and their interactions, as shown in Figure 1. Since the deformation of the droplet occurs only in the bottom part of the air domain in the *XY* plane view, the simulation of a droplet in an entire annular groove and half of the groove was first performed. The initial conditions, boundary conditions, and material properties of the two models were the same, and the acceleration was given as 20 m/s^2^. The results after 0.3 s time step are shown in Figure 2.

The characteristic size of the droplet deformation in the entire annular groove and half annular groove is shown in Table 1.

The deformation of the droplet in the entire annular groove is the same as that of a half annular groove. Hence, a half annular channel was used for the air domain as equivalent to the complete annular channel in numerical simulations, as illustrated in Figure 3.

For the module of incompressible laminar flow, the initial condition of the velocity vector is 0 m/s in each direction, the pressure is 0 Pa, and the temperature is 20 °C.

For the module of LS, the LS function of the droplet domain is 1, and the air function is 0. Besides, the surface tension at the interface of the two-phases is considered.

The initial position of the droplet is at the bottom surface of the annular groove.

The simulation was carried out using both the incompressible laminar flow module and the LS two-phase flow module. The boundary conditions of the model are shown in Table 2:

Table 3 shows the properties of materials.

Phase transition mainly occurs inside the droplet and at the boundary surfaces between the droplet and air. Therefore, a special mesh refinement process was required for these regions. A mesh consisting of 96,976 elements was generated using the free tetrahedral meshing method, and the average cell quality of the mesh reached 0.6.

## 3. Result and Discussion

### 3.1. Characteristic Sizes of the Droplet Deformation

The characteristic-height (named C-height) and the characteristic-length (named C-length) of the droplet after deformation were recorded, as shown in Figure 4. The C-height is the distance between the highest point and the lowest point of the droplet in the *XY* plane, and the C-length is the contact length of the droplet and the upper surface in the *XZ* view. Besides, Figure 4 shows that when acceleration is 400 m/s^2^, the droplets will flatten at 0.1 s, 0.2 s, and 0.3 s. It can be seen from the *XZ* view that the flattened droplet expands and reaches the upper surface of the annular groove.

Here, the steady-state deformation of the droplet is the desired result, so that the relationship between the deformation of the droplet and the time step needs to be investigated. The transient performance of the step size from 0–0.3 s are simulated using the models with four accelerations of 100 m/s^2^, 200 m/s^2^, 300 m/s^2^, and 400 m/s^2^. Figure 5 shows the relationship between the time step and the C-height of the droplets.

The results show that the C-height of the droplet from the time steps of 0.01 s to 0.18 s will increase and decrease periodically, which indicates that the droplet is flattened after the initial acceleration, and rebounds with an increased time step. This oscillation in height repeats for half a second before it settles. Acceleration is always present and does not change with time, but the deformation of the droplet oscillates at the beginning. When the time step exceeds 0.18 s, the height change of the droplets gradually becomes stable, which means the influence of the time step on the C-height gradually fades away. Therefore, the results after 0.3 s are used in the study.

More types of acceleration were applied to study the deformation of the droplet and verify whether the droplet will split at a large acceleration. The acceleration varies from 10 to 600 m/s^2^, and several characteristic results at accelerations 30, 300, 500, and 600 m/s^2^ are shown in Figure 6.

Figure 7 shows the relationship between the C-height of the droplet deformation and the acceleration. The droplet in the annular groove is deformed by the acceleration, and the C-height of the droplet decreases as the acceleration increases. The C-height of the droplet is 1.658 mm at 10 m/s^2^, and 0.246 mm at 600 m/s^2^. Besides, it is observed that even when the acceleration reaches 600 m/s^2^, the droplet remained in one piece and did not break.

The simulation results also show that as the acceleration increases, the droplet is gradually flattened and the C-length increases. The relationship between acceleration and C-length is shown in Figure 8 (the red line represents the normalized data of C-length).

When the acceleration is lower than 50 m/s^2^, the little deformation is not enough for the droplet to be in contact with the upper surface of the annular groove. At this time, the C-length of the deformation is 0. When the acceleration reaches 60 m/s^2^, the droplet is deformed enough to contact the upper surface of the groove, and the C-length at this time is 0.863 mm. The length of the droplet gradually increases as the acceleration increases. When the acceleration is 600 m/s^2^, the C-length reaches 2.776 mm.

### 3.2. Effect of Material Properties in the Domain around the Droplet on Characteristic Sizes of the Deformation

In this section, the material properties surrounding the domain of the droplet are simulated. Figure 9a–c respectively show the deformation of the droplet as the viscosity, density, and surface tension of the annular region increase. Meanwhile, Figure 9d shows the deformation of the droplet when the material of the annular groove domain is water. Here, the acceleration is 300 m/s^2^, the viscosity is four times larger than air, (7.16 × 10^−5^ Pa.s), the density is three times bigger than air (3.9 kg/m^3^), and the surface tension is 1.44.

The relationship between the deformation characteristic size and acceleration of the droplet in different materials property domains is shown in Table 4.

It can be seen from Table 4 that the material properties of the domain surrounding the droplet affect the characteristic size of the deformation. Increasing the viscosity and density of the material can increase the C-height of deformation and decrease the C-length of deformation. However, density and viscosity have little effect on the characteristic size of deformation. Besides, the effect of surface tension on the characteristic sizes of deformation is significant. Increasing the surface tension will increase the C-height, but the change of C-length is not obvious. The water domain has a higher density and higher viscosity than the air domain. Hence, the C-height of the mercury droplet in water is small and the C-length is large, which means that mercury is not easily deformed in water.

### 3.3. Experimental Setting and Results

The experiments for verifying the deformation of an accelerating droplet in the annular groove were designed, as shown in Figure 10. The setting of the experiment consisted of a turntable, a silicon wafer with a SU-8 photoresist annular groove (the size of which was 2.5mm × 2.5mm) attached to the top surface of the turntable, and a glass slide as the lid of the groove [23]. A mercury droplet of 2 mm in diameter was placed in the groove and surrounded by air. The contact angles between the droplet and the outer and lower surface in the annular groove surface and the glass substrate were 145°, 135°, and 150°, respectively. The deformation of the droplet was captured by a high-speed camera placed vertically above the turntable. The whole experiment bench was placed in the room temperature of 20 °C.

The centripetal acceleration is generated using the turntable. The distance between the initial position of the droplet and the center of the turntable is 20 mm, and different additional accelerations can be obtained by changing the speed of the turntable. The acceleration can be calculated with the following Equation:(13)$$a=\omega^{2}r$$
where *a* is the acceleration, *ω* is the angular velocity of turntable, and *r* is the distance between the initial position of the droplet and the center of the turntable.

Since the steady-state deformation model was required, the image of the droplet was acquired 60 s after the speed of the turntable became steady. Figure 11 shows the shape of the droplet when the rotation speed reached 440, 1050, 1420, and 1710 rpm. These speeds correspond to accelerations of 42, 242, 438, and 640 m/s^2^, respectively.

For comparison of the C-height in the experimental results with the C-height in the simulation results, the boundary conditions in the simulation needed to be improved. The modified boundary conditions, including the contact angle, are shown in Table 5:

The wetted wall was added as a boundary condition in the Multiphysics coupling, reflecting the contact situation between the droplet, and the annular groove and other initial conditions and material properties remain unchanged in simulation. The simulation and experimental results of the droplet deformation when the accelerations were 42, 242, 438, and 640 m/s^2^ are shown in Table 6.

Figure 12a shows the relationship between the C-height of the droplet deformation and the acceleration. The black line represents the simulation results, and the red dots represent the experimental results. In addition, Figure 12b shows the relationship between the C-length of the deformation and the acceleration, and the red line represents the normalized data of C-length.

By measuring the C-height of the droplet in Figure 11 and calculating the C-height in the simulation models, the experimental results and simulation results of the C-height and acceleration are plotted shown in Figure 12a. For the simulation models, the condition of adding the contact angles (wetted wall) will also cause the droplet to be squeezed under acceleration. However, compared with the sliding wall boundary condition, the C-height of the model with the contact angles boundary condition is increased under the same acceleration, which indicates that the contact angles boundary condition will reduce the deformation of the droplet. The C-height of the droplet was decreasing with the acceleration increases, as shown in Figure 12a, and the droplet was still only flattened and did not break when the rotational speed reaches 1710 rpm. (The black line in Figure 12a represents the numerical simulation results, and the red dots represent the experimental results.) This shows the same trend as the results of the simulation work. The relative error between the simulation and experimental results is 0.73% at 42 m/s^2^, 4.72% at 242 m/s^2^, 2.05% at 438 m/s^2^, and 11% at 640m/s^2^. The avenge amount of the relative error between the simulation and experimental results is 4.26%. Figure 12b shows the simulation results of the relationship between acceleration and C-length with contact angle boundary (wetted wall boundary) conditions models. The droplet started to contact the upper surface of the annular groove at an acceleration of 80 m/s^2^, and the C-length at this time was 1.148 mm. As the acceleration increases, the value of the C-length increases as the liquid is squeezed. When the acceleration value reaches 100 m/s^2^, the increasing trend of C-length gradually slows down and stabilizes at about 2.24 mm. The results show that considering the contact angle boundary (wetted wall boundary) condition will increase the droplet C-height and decrease the C-length, which represents a reduction in the deformation amount of the droplet.

## 4. Conclusions

This paper presents numerical and experimental research of a liquid metallic droplet deformation in an annular groove channel under the influence of accelerations. The Laminar LS method was used in the numerical simulation. The simulation model consists of two phases: the air phase of a half annular groove of 2.5 × 2.5 mm in cross-section, and the mercury liquid phase with a diameter of 2 mm. Accelerations were applied to the fluid in a direction perpendicular to the direction of gravitational acceleration. Initially, the droplet is resting in the groove and only in contact with the bottom of the groove. The characteristic size of the droplet after deformation was proposed to show how acceleration will deform the droplet, and the deformation of a mercury droplet in different material parameter domains was analyzed. In addition, the experimental results were used to verify the results of the simulation work. An experimental setting was designed to match the simulation model. A turntable was used to generate centripetal acceleration in the experiment work as the acceleration in the simulation model, and a high-speed camera was used to capture the image of droplet deformation.

The numerical simulation results show that the C-height of the droplet decreases as the acceleration increases. At 10 m/s2, the height of the droplet was 1.658 mm, and at 600 m/s^2^, it was reduced to 0.246 mm. Meanwhile, the C-length of the droplet increases with the acceleration increase. The length was 0.863 mm at acceleration 60 m/s^2^ and increased to 2.776 mm at 600 m/s^2^. Besides, the simulation results show that the deformation of the droplet is different in domains of different material properties. A larger viscosity, density, and surface tension will cause the reduction of the droplet deformation. Hence, the droplet is less deformed in water than air. The simulation results also show that droplet deformation requires initial oscillations of <0.18s. Therefore, for the droplet micro-switches and droplet acceleration sensors, the initial oscillation of the droplet under acceleration needs to be considered during the sensor design process. Appropriately reducing the deformation requirements can eliminate the effect of initial oscillation and further improve the sensitivity. The experimental results also show that droplet will be squeezed under the effect of accelerations. Both the experimental and simulation results show the same droplet deformation trend. This research work contributes to the research community of liquid droplet sensors in the following aspects: firstly, for the droplet inclination sensors, which require a small deformation of the droplet when affect by acceleration, the material around the metal droplet can be changed to reduce the deformation of the droplets when affected by acceleration; secondly, for the droplet acceleration sensors, which use the principle of droplet deformation under the action of acceleration to output an acceleration signal, the structures of the droplet acceleration sensor can be reasonably designed by the C-length and C-height results of the droplet.

## Figures and Tables

**Figure 1 biosensors-10-00061-f001:**
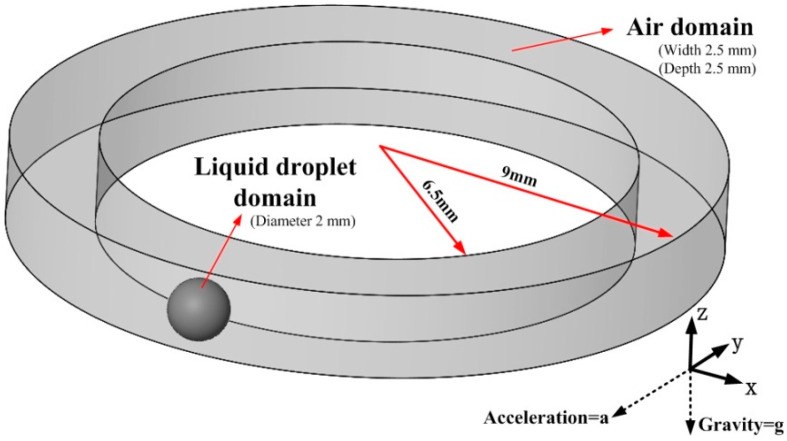
Schematic diagram of the 3D model. The outer and inner diameters of the groove are 18 mm and 13 mm, respectively. The direction of acceleration is in negative Y and gravity in negative Z.

**Figure 2 biosensors-10-00061-f002:**
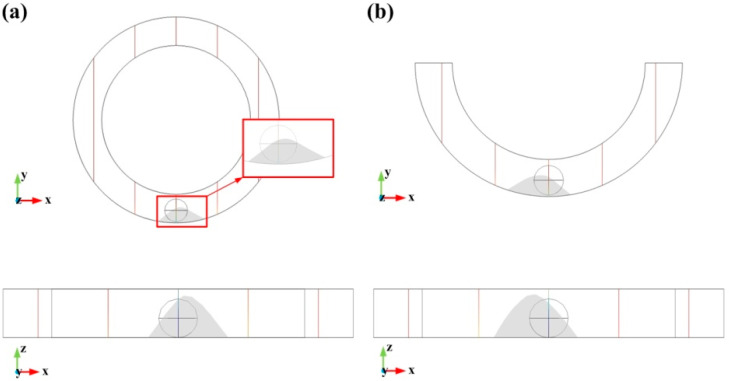
Results of the droplet deformation after 0.3 s time step: (**a**) is the entire annular groove result; (**b**) is the half annular groove result.

**Figure 3 biosensors-10-00061-f003:**
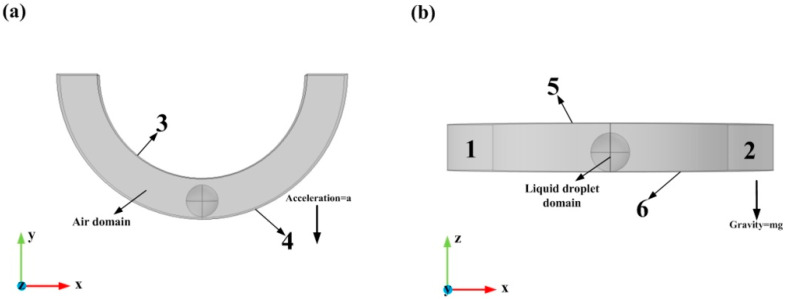
3D model, initial, and boundary condition of the liquid droplet domain and the air domain: (**a**) is the top view of the model; (**b**) is the front view of the model.

**Figure 4 biosensors-10-00061-f004:**
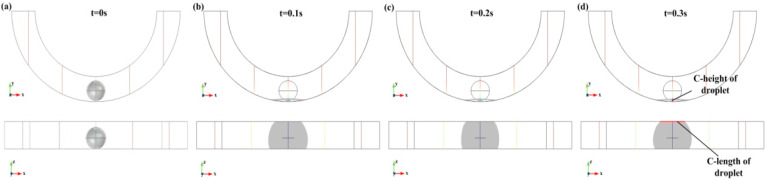
The images of droplet deformation after 0.3 s: (**a**–**d**) are the droplet deformation results with time steps of 0 s, 0.1 s, 0.2 s, and 0.3 s, respectively.

**Figure 5 biosensors-10-00061-f005:**
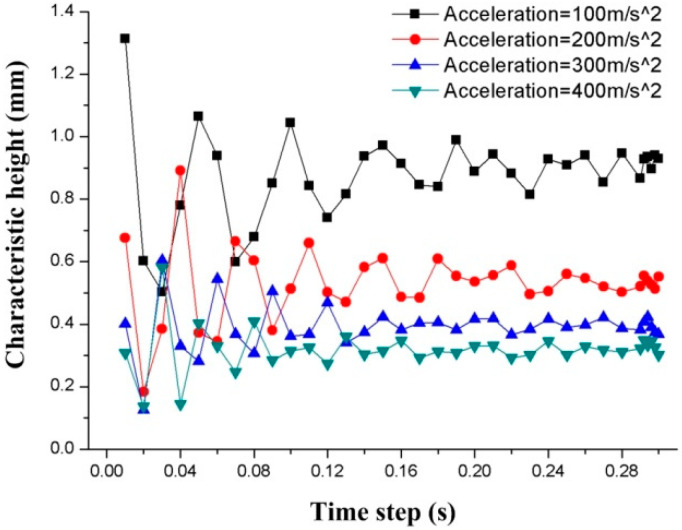
The relationship between characteristic-height (C-height) and the time step in different accelerations. The accelerations are from 100 m/s^2^ to 400 m/s^2^.

**Figure 6 biosensors-10-00061-f006:**
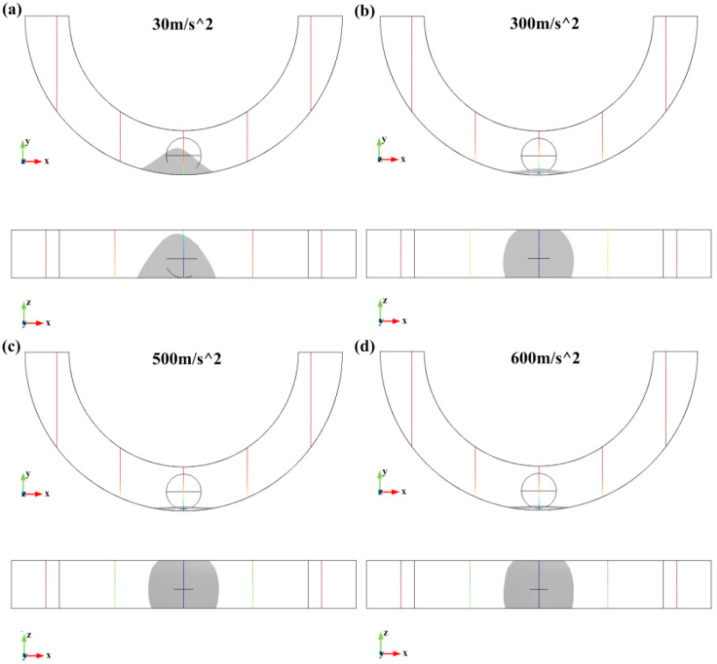
The deformations of the droplet at accelerations (**a**) 30, (**b**) 300, (**c**) 500, and (**d**) 600 m/s^2^, respectively.

**Figure 7 biosensors-10-00061-f007:**
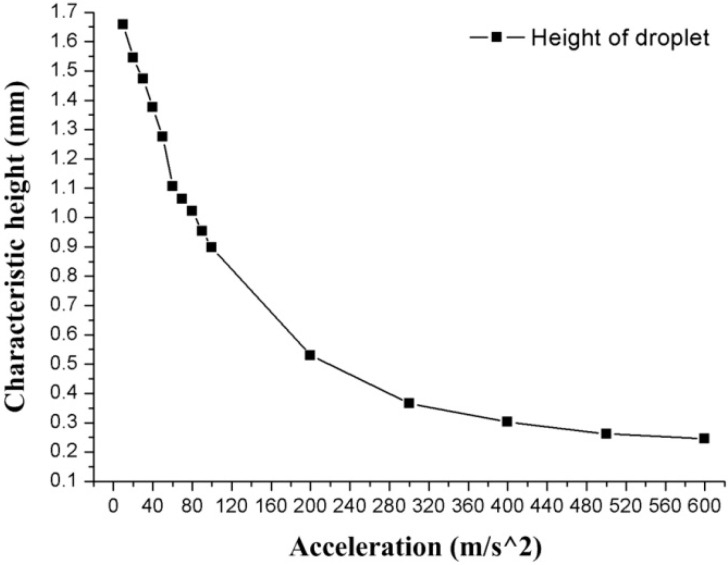
The relationship between the C-height and the acceleration.

**Figure 8 biosensors-10-00061-f008:**
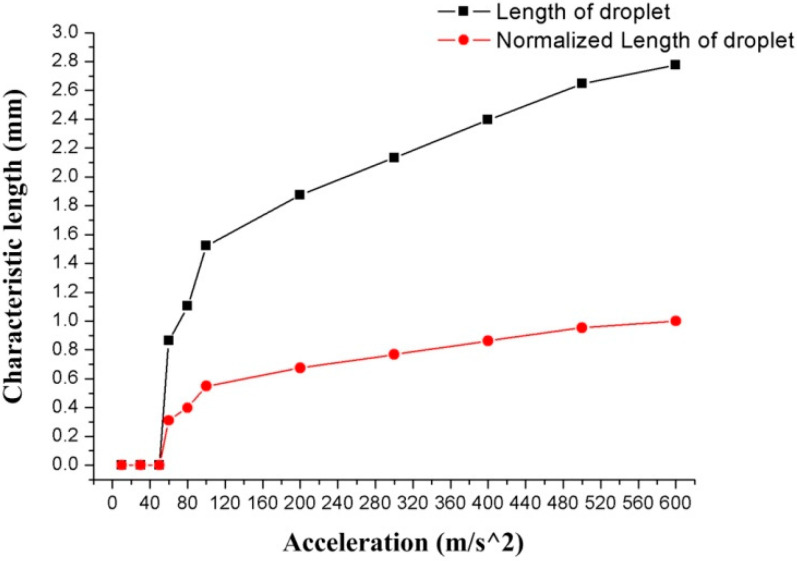
The results of characteristic-length (C-length) of droplet deformation at different accelerations.

**Figure 9 biosensors-10-00061-f009:**
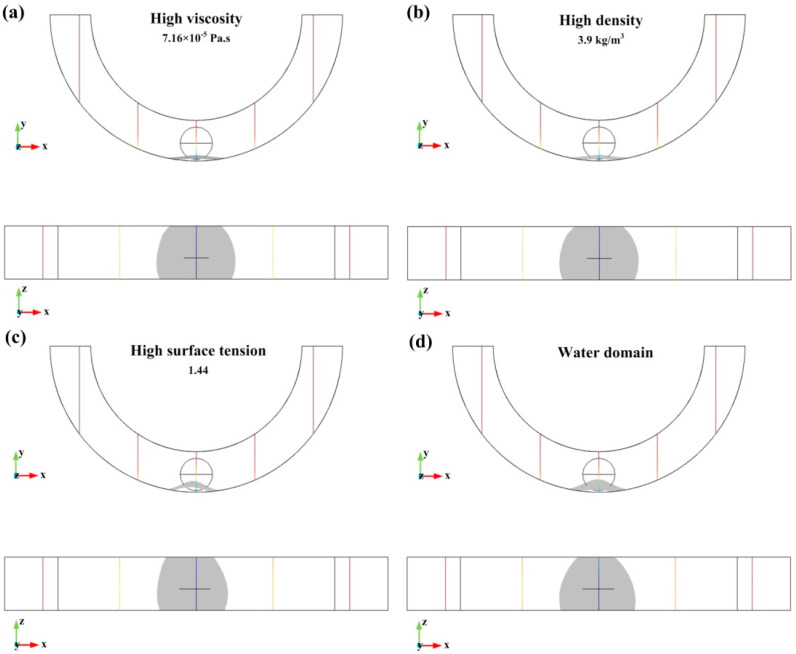
The simulation results of the droplet deformation in different material properties; (**a**–**d**) are the deformation of acceleration in high viscosity, high density, high surface tension, and water domain, respectively.

**Figure 10 biosensors-10-00061-f010:**
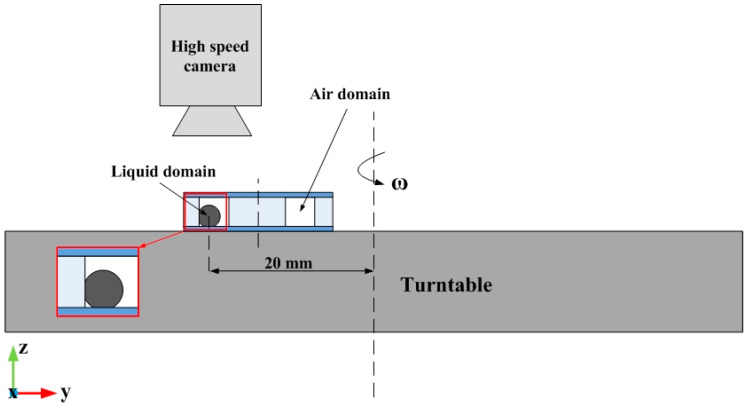
Schematic of the experimental workbench consisting of a turntable, a structure with groove and droplet, and a high-speed camera.

**Figure 11 biosensors-10-00061-f011:**
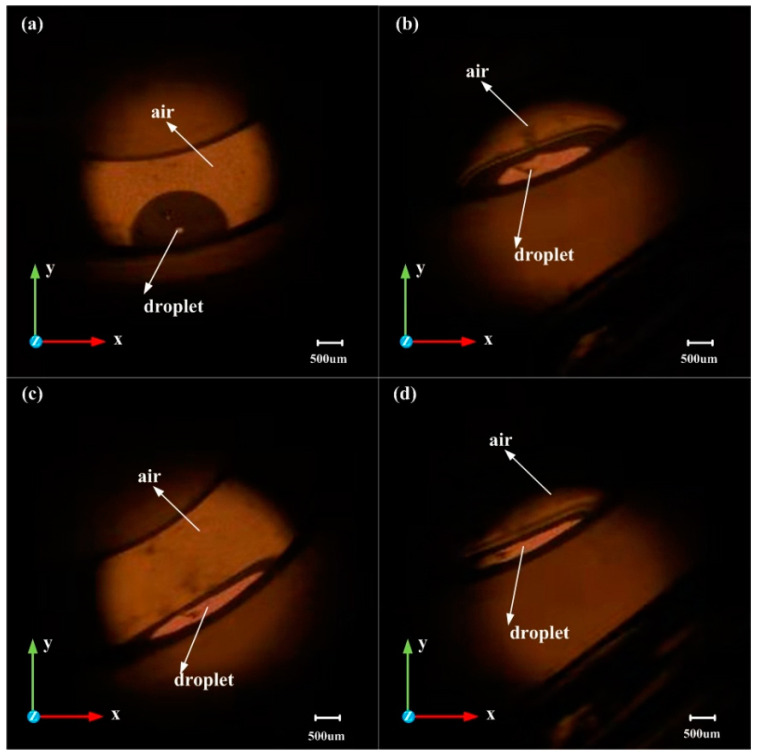
The shape of droplet in the *XY* plane: (**a**) at 42 m/s^2^, (**b**) at 242 m/s^2^, (**c**) at 438 m/s^2^, and (**d**) at 640 m/s^2^.

**Figure 12 biosensors-10-00061-f012:**
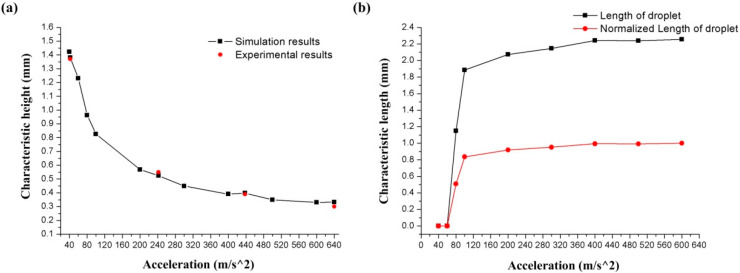
The characteristic sizes of the droplet deformation in different acceleration: (**a**) shows the results of C-height; (**b**) shows the results of C-length.

**Table 1 biosensors-10-00061-t001:** Characteristic size of the droplet in the entire annular groove and half annular groove.

	C-Height (mm)	C-Length (mm)
Entire annular groove	1.548	0
Half annular groove	1.544	0

**Table 2 biosensors-10-00061-t002:** The boundary conditions of the model.

	Module	
**Boundary**	Laminar flow	Level set
1, 2, 3	No slipping wall	Not active
4,5,6	Slipping wall	

**Table 3 biosensors-10-00061-t003:** The properties of materials.

	Materials	
**Properties**	Mercury	Air
Density (kg/m^3^)	13,600	1.3
Viscosity (Pa.s)	1.526 × 10^−3^	1.79 × 10^−5^
Surface tension (N/m)	0.48	

**Table 4 biosensors-10-00061-t004:** The characteristic sizes of deformation in different material properties.

	High Viscosity	High Density	High Surface Tension	Water Domain	Air Domain
C-height (mm)	0.379	0.362	0.681	0.798	**0.345**
C-length (mm)	2.228	2.231	2.217	1.895	**2.244**

**Table 5 biosensors-10-00061-t005:** The boundary conditions of the model.

	Module		
Boundary	Laminar Flow	Level Set	Multiphysics Coupling
1, 2, 3	No slipping wall	Not active	
4	Not active	Not active	Wetted wall (contact angle 145°)
5			Wetted wall (contact angle 135°)
6			Wetted wall (contact angle 150°)

**Table 6 biosensors-10-00061-t006:** Simulation and experimental C-height of droplet.

	42 m/s^2^	242 m/s^2^	438 m/s^2^	640 m/s^2^
Simulation height (mm)	1.380	0.524	0.398	0.333
Experimental height (mm)	1.37	0.55	0.39	0.30

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
