# Peer review of "A Deformation of a Mercury Droplet under Acceleration in an Annular Groove"

_biosensors, 2020, doi:10.3390/bios10060061_

Round 1

Reviewer 1 Report

Deformation of a mercury droplet under acceleration in an annular groove

In this manuscript, the authors performed a numerical and experimental study of the deformation of a mercury droplet in an annular groove at acceleration. Laminar LS method which was used in their numerical enables tracking the profile of droplets during the deformation process. The relationship between the height and length of the droplet, and acceleration was obtained. The results of simulation and experiments both show that the droplet deforms under acceleration, and the characteristic height of droplet decreases with the increase of acceleration.

The manuscript is well written and organized. This work is valuable for publishing in Biosensors Journal, however, minor revisions and considerations are necessary before publishing.

  1. The authors mentioned that “the droplet does not split even when the acceleration is up to 600 m/s2”. Based on their experimental analysis, at what acceleration the droplet splitting occurs?
  2. As the viscosity, density, and surface tension of the annular region increase, the deformation of the droplet has been studied. Could the authors comment on the deformation of the droplet as a function of acceleration as the viscosity, density, and surface tension of droplet change? Could the authors comment on the circumstances that result in droplet splitting?
  3. In section 3.3, it is mentioned that “the surface of annular groove was superhydrophobic”. First, how did the authors obtain the superhydrophobicity on these surfaces (what is the chemistry/surface texture?). Second, the contact angles are obtained by the studied droplet, mercury (“The contact angles between the droplet and the outer and lower surface …”), so what are the contact angles with water? You can only claim the superhydrophobicity of a surface if it is characterized by water contact angle not mercury which has such a high surface tension.
  4. There are a few missing parts in the text that need to be revisited: (a) all the used terms in equations have to be defined, this is not fulfilled with equation 9 in the manuscript. (b) the normalized length of the droplet graph/data shown in Figure 8 has not been pointed out in the text.

Author Response

In this manuscript, the authors performed a numerical and experimental study of the deformation of a mercury droplet in an annular groove at acceleration. Laminar LS method which was used in their numerical enables tracking the profile of droplets during the deformation process. The relationship between the height and length of the droplet, and acceleration was obtained. The results of simulation and experiments both show that the droplet deforms under acceleration, and the characteristic height of droplet decreases with the increase of acceleration.

The manuscript is well written and organized. This work is valuable for publishing in Biosensors Journal, however, minor revisions and considerations are necessary before publishing.

  1. The authors mentioned that “the droplet does not split even when the acceleration is up to 600 m/s2”. Based on their experimental analysis, at what acceleration the droplet splitting occurs?

   We did an experiment by increasing the rotation speed of the turntable to 2110 rpm and increasing the distance between the droplet and the turntable center from 20 mm to 30 mm. (The rest of the conditions are unchanged and the acceleration is 1463m/s2) The deformation of the droplet is shown in Figure. No breakage of the droplet was observed.

  1. As the viscosity, density, and surface tension of the annular region increase, the deformation of the droplet has been studied. Could the authors comment on the deformation of the droplet as a function of acceleration as the viscosity, density, and surface tension of droplet change? Could the authors comment on the circumstances that result in droplet splitting?

Equation (1), (2) are laminar flow formula which can calculate the velocity distribution and pressure distribution in the fluid under acceleration. The equation includes density, viscosity, and surface tension. Besides, Equation (4) is Level Set formula that can be used to track the boundary of two phases. The Level Set equation corrects the density and viscosity of materials. The deformation profile of mercury can be calculated by Equation (1,2,4).

The experimental results show that after increasing the acceleration to 1463m/s2, the droplet is still squashed without breaking up. Combined with the results in reference[14]. The droplet breakup will occur when a droplet with a relatively high initial velocity and hits a flat plate under the action of acceleration. Besides, the droplets staying on the flat plate will only deform under the action of uniform acceleration and are difficult to split.

  1. In section 3.3, it is mentioned that “the surface of annular groove was superhydrophobic”. First, how did the authors obtain the superhydrophobicity on these surfaces (what is the chemistry/surface texture?). Second, the contact angles are obtained by the studied droplet, mercury (“The contact angles between the droplet and the outer and lower surface …”), so what are the contact angles with water? You can only claim the superhydrophobicity of a surface if it is characterized by water contact angle not mercury which has such a high surface tension.

   The description of the superhydrophobic surface is due to the high surface tension of the mercury droplet not the Hydrophobic treatment of the surface. Therefore, in the modified manuscript, the description of the superhydrophobic surface of the annular groove is deleted, and only the contact angle of the mercury droplet with the three surfaces is introduced.

  1. There are a few missing parts in the text that need to be revisited: (a) all the used terms in equations have to be defined, this is not fulfilled with equation 9 in the manuscript. (b) the normalized length of the droplet graph/data shown in Figure 8 has not been pointed out in the text.

    We modified the description of Equation 9

                                 (9)

where σ is surface tension coefficient, δ(Ï•) is Dirac delta function.

We modified the description of normalized length of the droplet graph/data shown in Figure 8: “The simulation results also show that as the acceleration increases, the droplet is gradually flattened and the C-length increases. The relationship between acceleration and C-length is shown in figure 8. (The red line represents the normalized data of C-length).”

Reviewer 2 Report

1: The authors have used a mercury droplet in an annular grove to establish its application as a liquid MEMs acceleration sensor. The simulation and experimental research design approach seems logical and follows the standard research structure to support their key conclusion- droplet heigh reduces under acceleration. The authors have also qualitatively established a relation between the variable parameters and the droplet height/length which they have studied. I do have few minor suggestions/comments as follows: 2: The authors have not addressed as to why the size of 2.5 mm by 2.5 mm was chosen or why was only mercury chosen as the studied liquid droplet, does it follow any standard sensor currently available ? 3: The authors have listed a bunch of references that support the Level Set method to study such sensors, but the authors in the manuscript have not made clear as to what is the motivation behind using liquid droplet modelling ? Has this not been studied before ? Why this technique needs to be studied, is it useful for new generation of MEMS based sensors ? The authors need to establish the need/application for their chosen study in the introduction section more clearly. 4: The authors have qualitatively concluded that droplet height will decrease with acceleration, but how ill that affect the overall sensing capabilities in a system has not been quantified or eben referred to ? The authors start by addressing liquid sensors but have not concluded at all in the manuscript as to how this study can be applied to sensing ? 5: How was the Reynolds number determined to be less than 150 ? 6: In the simulation, the authors have tried to include variation by changing surrounding fluid and other parameters, can the authors also support that with experimental data ? Also, why various other boundary conditions such as coefficient of friction other than surface tension have not been considered ? 7: The authors need to explain the initial oscillations in the time domain of <0.18 sec more clearly and how will that impact the overall sensitivity of the sensor ? 8: Why superhydrophobic surface was considered in the simulation ? Any reference that supports that assumption ?

Author Response

1: The authors have used a mercury droplet in an annular grove to establish its application as a liquid MEMs acceleration sensor. The simulation and experimental research design approach seems logical and follows the standard research structure to support their key conclusion- droplet heigh reduces under acceleration. The authors have also qualitatively established a relation between the variable parameters and the droplet height/length which they have studied. I do have few minor suggestions/comments as follows:

2: The authors have not addressed as to why the size of 2.5 mm by 2.5 mm was chosen or why was only mercury chosen as the studied liquid droplet, does it follow any standard sensor currently available ?

In the manuscript, the deformation of mercury droplets under acceleration was simulated, and the diameter of the droplets was 2 mm. The size of the corresponding annular groove which controls the movement of the droplet was 2.5mm * 2.5mm. This manuscript is aimed at the study of droplet deformation in the MEMS droplet inclinometer sensor, and the structure is similar to the previous study by the author [3]. Besides, the size of the mercury droplets in the droplet acceleration/inclinometer sensor proposed by Park is larger than the motion groove. (Park U, Yoo K, Kim J. Development of a MEMS digital accelerometer (MDA) using a microscale liquid metal droplet in a microstructured photosensitive glass channel[J]. Sensors and Actuators A: Physical, 2010, 159(1): 51-57.) The sensor will be squashed in the groove during the working process, which reduces the resolution of sensors. Hence, the annular groove size was larger than the droplet diameter to leave gaps for the droplet. Meanwhile, increasing the size of the droplet can make the droplet have a smaller sliding angle, and improving the sensitivity of the sensor. Therefore, 2mm diameter droplets were selected in the study.

We modified the third paragraph of introduction section: “This paper presents a numerical and experimental study of the deformation of a mercury droplet in an annular groove at acceleration. The mercury was selected as the research object mainly because the high surface tension and conductivity of the mercury. Besides, mercury is widely used in current MEMS liquid droplet sensors.”

We modified the first paragraph of 2.1 section: “The material of the liquid droplet is mercury, which measures 2 mm in diameter. Here, the size of the annular groove is larger than the diameter of the droplet which makes the droplet easy to slide and improves the resolution of the sensor. The gravity is applied to the fluid in negative Z-direction, and the acceleration in negative Y-direction.”

3: The authors have listed a bunch of references that support the Level Set method to study such sensors, but the authors in the manuscript have not made clear as to what is the motivation behind using liquid droplet modelling? Has this not been studied before ? Why this technique needs to be studied, is it useful for new generation of MEMS based sensors ? The authors need to establish the need/application for their chosen study in the introduction section more clearly.

As a new type of sensor, the MEMS liquid droplet sensor currently has broad prospects in applications such as mobile phone communication equipment and automobiles. Due to the high viscosity of the metal droplets, the researchers claim that this type of sensor has a strong resistance to overload. However, there is no research on the deformation degree of the sensitive element in the droplet sensor under large acceleration. This manuscript analyzes the deformation of the droplet under large acceleration according to the author's previous research model. [3]

Establish the need/application of this work in the second paragraph of introduction section: “Aboutalebi used numerical simulation to study the influence of magnetic field intensity and droplet size on the ferromagnetic droplet splitting process in a symmetrical T-junction, and the boundary between the split and non-split regions where the droplets flow in the micro-T junction is predicted [15]. However, the research on the deformation of the droplet in an annular groove under a large acceleration is rare. Whether the deformation of the droplet in the annular groove is split or flattened, and the relationship between the deformation and the initial boundary or material is worth to be investigated.”

4: The authors have qualitatively concluded that droplet height will decrease with acceleration, but how ill that affect the overall sensing capabilities in a system has not been quantified or eben referred to ? The authors start by addressing liquid sensors but have not concluded at all in the manuscript as to how this study can be applied to sensing ?

The research results of the manuscript show that the mercury droplet is compressed and deformed by the acceleration in the annular groove and the C-height of the droplet becomes smaller and the C-length becomes larger when the acceleration increase. Besides, the research results also show that the deformation of the droplets is related to the viscosity, density, and surface tension of the material.

The research results of this manuscript provide a theoretical basis and design ideas for the application of droplet acceleration sensors and actuators (such as droplet switches). In addition, for the other droplet sensors, such as angle sensors, the manuscript provides an analysis of the deformation of sensitive components when the droplet is affected by external acceleration.

In the modified manuscript, we added some applications of the research results on the sensor in the conclusion section, including sensor material selection and structural design. “This research work contributes to the research community of liquid droplet sensors in the following aspects: First, for the droplet inclination sensors, which require a small deformation of the droplet when affect by acceleration, the material around the metal droplet can be changed to reduce the deformation of the droplets when affected by acceleration. Secondly, for the droplet acceleration sensors which use the principle of droplet deformation under the action of acceleration to output an acceleration signal, the structures of the droplet acceleration sensor can be reasonably designed by the C-length and C-height results of the droplet. ”

5: How was the Reynolds number determined to be less than 150 ?

First, as shown in the figure, the distribution of the unit Reynolds number after the droplet is deformed under the acceleration of 600m/s2. It can be seen that the maximum Reynolds number in the figure is only 140, and it has not reached 150, so the model belongs to laminar flow.

Secondly, as can be seen from reference [16,17], the model of droplets colliding with the plate is using the Laminar flow method. Since this model is similar to it, it is also Laminar flow when analyzing here (Reynolds number is less than 150).

6: In the simulation, the authors have tried to include variation by changing surrounding fluid and other parameters, can the authors also support that with experimental data ? Also, why various other boundary conditions such as coefficient of friction other than surface tension have not been considered ?

As shown in the figure, we try to inject water into the annular groove to study the deformation of the droplet after the acceleration. However, after the water is injected into the annular groove, a glass slide needs to be covered to complete the packaging step. The contact angle between the glass slide and water is small, and the water will adhere to the surface of the glass and flow away, which causes bubbles in the annular groove. Besides, the glass slide is combined with SU-8 annular groove using UV adhesive. This makes the overall structure unsealed. As the rotation speed of the turntable increases, the water in the annular groove will be thrown out (mercury droplets have not been thrown out of the annular groove due to the large surface tension and no cracking). Compared to water, mercury droplets tend to stay in the air. Therefore, no deformation results of mercury droplets in water can be observed due to the presence of bubbles. In addition, the overall structure is not sealed, so it is impossible to study the gas of other materials in the annular groove.

The boundary conditions of mercury droplets deformed by acceleration include surface tension and volume force. According to the reference[16,17], for the flow and deformation of the liquid, the friction coefficient has a smaller effect on the deformation of the droplet than the surface tension. Therefore, the friction coefficient is not considered in the model of droplet deformation under acceleration.

7: The authors need to explain the initial oscillations in the time domain of <0.18 sec more clearly and how will that impact the overall sensitivity of the sensor ?

The conclusion part was modified: “The simulation results also show that Droplet deformation requires initial oscillations of <0.18s. Therefore, for the droplet micro-switches and droplet acceleration sensors, the initial oscillation of the droplet under acceleration needs to be considered during the sensor design process. Appropriately reducing the deformation requirements can eliminate the effect of initial oscillation and further improve the sensitivity.”

8: Why superhydrophobic surface was considered in the simulation ? Any reference that supports that assumption ?

    The description of the superhydrophobic surface is due to the high surface tension of the mercury droplet not the Hydrophobic treatment of the surface. Therefore, in the modified manuscript, the description of the superhydrophobic surface of the annular groove is deleted, and only the contact angle of the mercury droplet with the three surfaces is introduced. Besides, there is a contact angle between the droplet and the surface of the annular groove, rather than ideal tangential contact.

Reviewer 3 Report

Comments to the authors:

General comments:

In section 3.2 it is simulated the effect of material properties, regarding viscosity, density, surface tension and water domain. But there are other effects not referred, for example temperature, that could be easily simulated and experimented. Moreover, those effects were not experimented, apart from the contact angle.

In 3.3 the experimental section, a figure with the photo and a cross-section design with dimensions of the turntable, a structure with groove and droplet should be given. You can introduce this information at supplementary material.

Are the simulations with the same conditions as the experimental? Same dimensions and same material inserted in the simulations? For comparison it is crucial. I noticed that the contact angle was revised for including the experimental in the simulations, but the other conditions?

“The relative error between the simulation and experimental results is 2.6% at 42 m/s2, 6.2% at 242 m/s2, and 4.0% at 438 m/s2.” How sure you can affirm that if you did not simulated for this specific accelerations?

Specific comments:

Page 3, section 2.2, “where u is the velocity vector, ρ is the density, and p is the pressure, and F is the volume” the word “and” before the p should be removed.

Page 3, section 2.2, “where g is the gravitational acceleration, α is the acceleration of the whole flow” the symbol alfa is different from the equation and from the text. Use the same symbol.

Page 4, “is the Heaviside function which is used to smooth”, introduce a comma before “which”

Page 5, “The deformation of the droplet in the entire annular groove is the same as that of a half annular groove.” You need to give more results for supporting this conclusion. Only by inspecting, visually, the figure a and b, we can not conclude that. You need to write some numerical results.

Page 7, “the droplet remained in one piece and did not break.” did you see when break?

Figure 9, the values of the materials properties changed and written in the last sentence of page 8, should be written at the caption of figure 9, as well as the acceleration value considered for that simulation.

Table 3: this table should have a comparison between the previous values and that ones for the same acceleration, as there is not a plot of these characteristics evolution. Because the reader must go back for comparing. Also, the values should be written in the table.

Page 9, “which means that mercury is not easily deformed in water.“, why? Possible explanation?

Page 9, “air. The surface of the annular groove was super-hydrophobic. The contact..” Again, why? explanation, due to the material?

Page 10: “Figure 11 shows the shape of the droplet when the rotation speed reached 440, 1050, 1420, and 1710 rpm. These speeds correspond to accelerations of 42, 242, 438, and 640 m/s2 respectively.” why those accelerations were not considered in the simulations?

Page 12, “In addition, Fig. 12(b) shows the relationship between the C-length of the deformation and the acceleration, and the red …” but for experimental or simulated?

Figure 12a - the simulation results should be for the same acceleration values.

Figure 12b - but for simulation or experimental?

Author Response

In section 3.2 it is simulated the effect of material properties, regarding viscosity, density, surface tension and water domain. But there are other effects not referred, for example temperature, that could be easily simulated and experimented. Moreover, those effects were not experimented, apart from the contact angle.

We did the simulation work of the droplet deformation in different temperatures. (The temperature are from 40℃to 100℃). The acceleration was 300 m/s2, and the rest boundary conditions and the material properties were the same. The results are shown in Figure.

Figure. Results of the droplet deformation in different temperatures

The C-height and C-length of the droplet deformation at different temperatures are shown in Table:

Table. C-height and C-length of the droplet deformation at different temperatures

Temperature

C-height (mm)

C-length (mm)

40℃

0.451

2.741

60℃

0.439

2.731

80℃

0.452

2.750

100℃

0.442

2.708

In Table, the C-height and C-length of the droplets do not change significantly with temperature. This is because the temperature is an indirect variable for the feature size of droplets. Equations 1 and 2 describe the velocity distribution and pressure distribution of the fluid when affected by acceleration. Viscosity, surface tension, and density are the direct variables that control the deformation of droplets. Changes in temperature will cause the above three variables to change. Besides, the surface tension and density of mercury decrease with increasing temperature, and the viscosity increase with increasing temperature. Therefore, studying the influence of temperature on the characteristic size of droplets does not reflect the deformation of droplets. So we did not add the simulation results of temperature to the manuscript.

As shown in the Figure, we try to inject water into the annular groove to study the deformation of the droplet after the acceleration. However, after the water is injected into the annular groove, a glass slide needs to be covered to complete the packaging step. The contact angle between the glass slide and water is small, and the water will adhere to the surface of the glass and flow away, which causes bubbles in the annular groove. Besides, the glass slide is combined with SU-8 annular groove using UV adhesive. This makes the overall structure unsealed. As the rotation speed of the turntable increases, the water in the annular groove will be thrown out (mercury droplets have not been thrown out of the annular groove due to the large surface tension and no cracking). Compared to water, mercury droplets tend to stay in the air. Therefore, no deformation results of mercury droplets in water can be observed due to the presence of bubbles. In addition, the overall structure is not sealed, so it is impossible to study the gas of other materials in the annular groove.

Figure. Inject water into the annular groove

In 3.3 the experimental section, a figure with the photo and a cross-section design with dimensions of the turntable, a structure with groove and droplet should be given. You can introduce this information at supplementary material.

We introduce the photo of the experimental bench, as shown in Figure.

Figure. Schematic diagram of experimental bench

Are the simulations with the same conditions as the experimental? Same dimensions and same material inserted in the simulations? For comparison it is crucial. I noticed that the contact angle was revised for including the experimental in the simulations, but the other conditions?

   The simulation conditions were the same as the experimental conditions. First, the material of droplet was mercury, and the droplet was surrounded by air. The diameter of droplet was 2mm and the size of annular groove was 2.5mm*2.5mm. Then, the temperature was 20℃. We modified the introduction of experimental conditions in the first paragraph of Section 3.3 as follows:

The experiments for verifying the deformation of an accelerating droplet in the annular groove were designed as shown in Figure 10. The setting of the experiment consists of a turntable, a silicon wafer with a SU-8 photoresist annular groove (The size of which is 2.5mm*2.5mm) attached to the top surface of the turntable and a glass slide as the lid of the groove. A mercury droplet of 2 mm in diameter was placed in the groove and surrounded by air.. The contact angles between the droplet and the outer and lower surface in the annular groove surface and the glass substrate were 145°, 135°, and 150° respectively. The deformation of the droplet was captured by a high speed camera placed vertically above the turntable. Besides, the whole experiment bench was placed in the room temperature of 20℃

“The relative error between the simulation and experimental results is 2.6% at 42 m/s2, 6.2% at 242 m/s2, and 4.0% at 438 m/s2.” How sure you can affirm that if you did not simulated for this specific accelerations?

Using the same boundary conditions and initial conditions, we did a supplementary simulation works of the droplet deformation at accelerations of 42 m/s2, 242 m/s2, 438 m/s2, and 640 m/s2. The results are shown in Table 6. Besides we modified the Figure 12(a) and added the simulation results of 42 m/s2, 242 m/s2, 438 m/s2, and 640 m/s2 in the figure.

Table 6. Simulation and experimental height of droplet

42 m/s2

242 m/s2

438 m/s2

640 m/s2

Simulation height(mm)

1.380

0.524

0.398

0.333

Experimental height(mm)

1.37

0.55

0.39

0.30

Figure 12. The characteristic sizes of the droplet deformation in different acceleration. (a) is the results of C-height, (b) is the results of C-length.

By comparing the simulation results and the experimental results, the relative errors are 0.73% at 42 m/s2, 4.72% at 242 m/s2, and 2.05% at 438 m/s2, 11% at 640 m/s2.

Specific comments:

Page 3, section 2.2, “where u is the velocity vector, ρ is the density, and p is the pressure, and F is the volume” the word “and” before the p should be removed.

We have removed the word “and” before the p.

Page 3, section 2.2, “where g is the gravitational acceleration, α is the acceleration of the whole flow” the symbol alfa is different from the equation and from the text. Use the same symbol.

We have modified the word “a” into “α” (alfa).

Page 4, “is the Heaviside function which is used to smooth”, introduce a comma before “which”

The modified sentence is: “the Heaviside function, which is used to smooth approximation and avoid the problem of non-convergence caused by the Gibbs phenomenon.”

Page 5, “The deformation of the droplet in the entire annular groove is the same as that of a half annular groove.” You need to give more results for supporting this conclusion. Only by inspecting, visually, the figure a and b, we can not conclude that. You need to write some numerical results.

We calculated the characteristic size of droplet deformation in the entire annular groove and the half annular groove, and the results are shown in Table 1.

Table 1. characteristic size of droplet in entire annular groove and half annular groove

C-height (mm)

C-length (mm)

Entire annular groove

1.548

0

Half annular groove

1.544

0

The deformation of the droplet in the entire annular groove is the same as that of a half annular groove.

Page 7, “the droplet remained in one piece and did not break.” did you see when break?

    We added an experiment by increasing the rotation speed of the turntable to 2110 rpm and increasing the distance between the droplet and the turntable center from 20 mm to 30 mm. (The rest of the conditions are unchanged and the acceleration is 1463m/s2) The deformation of the droplet is shown in Figure. No breakage of the droplet was observed.

Figure 9, the values of the materials properties changed and written in the last sentence of page 8, should be written at the caption of figure 9, as well as the acceleration value considered for that simulation.

    We modified the Fig.9, as shown below:

Figure 9. The simulation results of the droplet deformation in different material properties, (a)-(d) are the deformation of acceleration in high viscosity, high density, high surface tension, and water domain respectively

Table 3: this table should have a comparison between the previous values and that ones for the same acceleration, as there is not a plot of these characteristics evolution. Because the reader must go back for comparing. Also, the values should be written in the table.

    We modified Table 4 and added the air domain item as shown below. The bold items air domain is the deformation results when mercury droplet surrounded by air and with a 300 m/s2 of acceleration.

Table 4. The characteristic sizes of deformation in different material properties

High viscosity

High density

High surface tension

Water domain

Air domain

C-height (mm)

0.379

0.362

0.681

0.798

0.345

C-length (mm)

2.228

2.231

2.217

1.895

2.244

Page 9, “which means that mercury is not easily deformed in water.“, why? Possible explanation?

We have modified the expression of this paragraph as shown following:

“The water domain has higher density and higher viscosity than the air domain. Hence, the C-height of the mercury droplet in water is small and the C-length is large, which means that mercury is not easily deformed in water.”

The intuitive reason why droplets are not easily deformed in water is that the C-height of the mercury droplet in water is small and the C-length is large. The reason for this result is that the viscosity and density of water are greater than air.

Page 9, “air. The surface of the annular groove was super-hydrophobic. The contact..” Again, why? explanation, due to the material?

The difference in the contact angle of the droplet on the surface of the annular groove is caused by the different materials

The material of the lower substrate is silicon, (lower surface of annular groove) and the contact angle of the mercury on the silicon surface is 150°. The material of the outer surface of the annular groove is SU-8 photoresist, and the contact angle is 145°. The material of top surface is glass and the contact angle is 135°.

However, if the droplet is not mercury but water, the contact angle is not so large. Therefore, we have modified the description of the annular groove and deleted the description of super-hydrophobic.

Page 10: “Figure 11 shows the shape of the droplet when the rotation speed reached 440, 1050, 1420, and 1710 rpm. These speeds correspond to accelerations of 42, 242, 438, and 640 m/s2 respectively.” why those accelerations were not considered in the simulations?

We added the simulation works of the droplet deformation when the accelerations were 42, 242, 438, and 640 m/s2. The result of the droplet deformation is shown in Table 6

Table 6. Simulation and experimental C-height of droplet:

42 m/s2

242 m/s2

438 m/s2

640 m/s2

Simulation height(mm)

1.380

0.524

0.398

0.333

Experimental height(mm)

1.37

0.55

0.39

0.30

Page 12, “In addition, Fig. 12(b) shows the relationship between the C-length of the deformation and the acceleration, and the red …” but for experimental or simulated?

Figure 12a - the simulation results should be for the same acceleration values.

Figure 12b - but for simulation or experimental?

The black line in Fig. 12(a) represents the numerical simulation results and the red dots represent the experimental results.

Fig.12 (a) is modified and added the simulation results of droplet deformation when the accelerations were 42, 242, 438, and 640 m/s2 which are black dots in the curve.

Figure 12(b) shows the simulation results of the relationship between acceleration and C-length with contact angle boundary (wetted wall boundary) conditions models.

Round 2

Reviewer 3 Report

The authors did the changes and answer the review's comments, however, some of them should be better reflected in the manuscript:

  • The study of the influence of temperature on the characteristic size of droplets, despite not reflect the deformation of droplets should be given in supplementary material. You give the explanation for the reviewer, that explanation and table and values should be introduce in supplementary material. The readers should know these conditions.
  • You should introduce in supplementary material not only the experimental setup, but also a cross-section design with dimensions of the turntable, a structure with groove and droplet. It is not enough the setup. 
  • In section 3.3 when you refer fabrication you should introduce a reference to a SU-8 fabrication process. For example, "The setting of the experiment consists of a turntable, a silicon wafer with a SU-8 photoresist [introduce a Reference] annular groove. For the same Editor, MDPI in micromachines, in a special issue dedicated to SU-8 fabrication, a fabrication process similar to your, without cleanroom: "Micromachines 2014, 5(3), 738-755; https://doi.org/10.3390/mi5030738". You should introduce this reference.

Author Response

The authors did the changes and answer the review's comments, however, some of them should be better reflected in the manuscript:

  1. The study of the influence of temperature on the characteristic size of droplets, despite not reflect the deformation of droplets should be given in supplementary material. You give the explanation for the reviewer, that explanation and table and values should be introduce in supplementary material. The readers should know these conditions.

Response from authors: We added the study of the influence of temperature on the characteristic size of droplets in the supplementary material.

The simulation results of the droplet deformation in different temperatures are shown in the Figure below. (The temperature was from 40℃ to 100℃, and the acceleration was 300 m/s2, and the rest boundary conditions and the material properties were the same.)

Figure. Results of the droplet deformation in different temperatures.

The C-height and C-length of the droplet deformation at different temperatures are shown in Table:

Table. C-height and C-length of the droplet deformation at different temperatures

Temperature

C-height (mm)

C-length (mm)

40℃

0.451

2.741

60℃

0.439

2.731

80℃

0.452

2.750

100℃

0.442

2.708

The C-height and C-length of the droplets do not change significantly with temperature. This is because the temperature is an indirect variable for the feature size of droplets. Equations 1 and 2 describe the velocity distribution and pressure distribution of the fluid when affected by acceleration. Viscosity, surface tension, and density are the direct variables that control the deformation of droplets. Changes in temperature will cause the above three variables to change.

  1. You should introduce in supplementary material not only the experimental setup, but also a cross-section design with dimensions of the turntable, a structure with groove and droplet. It is not enough the setup.

Response from authors: We modified the schematic diagram of the experimental bench and added the detailed diagram of the turntable in the supplementary material section.

Figure. Schematic diagram of experimental bench. (a) is the overall schematic diagram of the experimental bench. (b) is the detailed diagram of the turntable.

  1. In section 3.3 when you refer fabrication you should introduce a reference to a SU-8 fabrication process. For example, "The setting of the experiment consists of a turntable, a silicon wafer with a SU-8 photoresist [introduce a Reference] annular groove. For the same Editor, MDPI in micromachines, in a special issue dedicated to SU-8 fabrication, a fabrication process similar to your, without cleanroom: "Micromachines 2014, 5(3), 738-755; https://doi.org/10.3390/mi5030738". You should introduce this reference.

Response: This paper is cited in the section 3.3 to introduce a reference of manufacture of annular groove.
